# *Caulis Polygoni Multiflori* Accelerates Megakaryopoiesis and Thrombopoiesis via Activating PI3K/Akt and MEK/ERK Signaling Pathways

**DOI:** 10.3390/ph15101204

**Published:** 2022-09-28

**Authors:** Xin Yang, Long Wang, Jing Zeng, Anguo Wu, Mi Qin, Min Wen, Ting Zhang, Wang Chen, Qibing Mei, Dalian Qin, Jing Yang, Yu Jiang, Jianming Wu

**Affiliations:** 1School of Pharmacy, Southwest Medical University, Luzhou 646000, China; 2School of Graduate, Southwest Medical University, Luzhou 646000, China; 3School of Basic Medical Sciences, Southwest Medical University, Luzhou 646000, China; 4Key Medical Laboratory of New Drug Discovery and Druggability Evaluation, Southwest Medical University, Luzhou 646000, China; 5Luzhou Key Laboratory of Activity Screening and Druggability Evaluation for Chinese Materia Medica, Southwest Medical University, Luzhou 646000, China

**Keywords:** *Caulis Polygoni Multiflori*, thrombocytopenia, PI3K/Akt, MEK/ERK, megakaryopoiesis, thrombopoiesis

## Abstract

Thrombocytopenia is one of the most common complications of cancer therapy. Until now, there are still no satisfactory medications to treat chemotherapy and radiation-induced thrombocytopenia (CIT and RIT, respectively). *Caulis Polygoni Multiflori* (CPM), one of the most commonly used Chinese herbs, has been well documented to nourish blood for tranquilizing the mind and treating anemia, suggesting its beneficial effect on hematopoiesis. However, it is unknown whether CPM can accelerate megakaryopoiesis and thrombopoiesis. Here, we employ a UHPLC Q–Exactive HF-X mass spectrometer (UHPLC QE HF-X MS) to identify 11 ingredients in CPM. Then, in vitro experiments showed that CPM significantly increased megakaryocyte (MK) differentiation and maturation but did not affect apoptosis and lactate dehydrogenase (LDH) release of K562 and Meg-01 cells. More importantly, animal experiments verified that CPM treatment markedly accelerated platelet recovery, megakaryopoiesis and thrombopoiesis in RIT mice without hepatic and renal toxicities in vivo. Finally, RNA-sequencing (RNA-seq) and western blot were used to determine that CPM increased the expression of proteins related to PI3K/Akt and MEK/ERK (MAPK) signaling pathways. On the contrary, blocking PI3K/Akt and MEK/ERK signaling pathways with their specific inhibitors suppressed MK differentiation induced by CPM. In conclusion, for the first time, our study demonstrates that CPM may be a promised thrombopoietic agent and provide an experimental basis for expanding clinical use.

## 1. Introduction

Cancer has become a leading cause of death globally; moreover, cancer morbidity and mortality continuously grow with population aging and environmental change [1]. It is reported that 19.3 million new cancer cases occur in 2020 and the total cancer population is expected to reach 28.4 million by 2040 [2]. Recently, a series of innovative treatment modalities, such as immunotherapy and targeted therapy, have had positive effects, but chemotherapy and radiotherapy remain un-substitutable in eradicating tumor cells sensitive to irradiation or as a preoperative adjuvant regimen [3]. Therefore, it should be noted that adverse reactions induced by chemoradiotherapy are unavoidable, partly due to its expanded irradiation field and great post-treatment necrosis [4]. Thrombocytopenia is the most frequent complication of cancer treatment, as evidenced by the low number of platelets (<10 × 10^10^ cells/L) [5], which can usually lead to treatment interruption and an increase in morbidity and mortality [6].

Platelets are specialized blood cells released from megakaryocytes (MKs). Megakaryopoiesis and thrombocytopoiesis are complex biological processes in which MKs are developed by multiple stages of lineage commitment of hematopoietic stem cells (HSCs), then undergo a complicated process, including multiple rounds of endomitosis, increased cytoplasm volume, formation of a demarcation membrane system (DMS), extended long branching to form proplatelet, and eventually leading to the release of platelets [7]. Thrombopoietin (TPO) and its receptor (c-Mpl) are crucial for MK differentiation and thrombopoiesis, which can activate several downstream signaling pathways, such as Janus kinase/signal transducers and activators of transcription (JAK/STAT), mitogen-activated protein kinase (MAPK), and phosphoinositol-3-kinase/Akt (PI3K/Akt). Additionally, other cytokines can regulate megakaryopoiesis and thrombocytopoiesis, such as granulocyte-macrophage colony stimulating factor (GM-CSF), interleukin (IL)-6 family and fibroblast growth factor (FGF) [8,9,10,11].

At present, conventional thrombocytopenia treatment strategies include reducing platelet consumption and boosting platelet generation by non-pharmacotherapies and pharmacotherapies [12,13]. Of the medications, cytokine medicines, such as recombinant human interleukin-11 (rhIL-11) and recombinant human thrombopoietin (rhTPO) [14,15]), and thrombopoietin receptor (TPOR) agonists approved by the FDA (including romiplostim, eltrombopag, avatrombopag and lusutrombopag [16,17,18]), have been proven to be effective thrombopenia treatments based on their potent thrombopoietic activities. Such drugs can increase peripheral blood (PB) platelet counts by facilitating HSC proliferation, MK growth and differentiation, as well as platelet production [14,15,16,17,18]. However, there are still no safe drugs to treat thrombocytopenia caused by chemoradiotherapy. rhIL-11, rhTPO and PEG-rHuMGDF, the first-generation thrombopoietic agents, have been reported to possess prospective efficacy on chemotherapy-induced thrombocytopenia (CIT). Unfortunately, these thrombopoietic agents contribute to a series of adverse events, including fever, arrhythmia, myalgia, edema or leading to neutralizing antibodies in clinical subjects, which ultimately leads to their discontinued application in Western countries [14,19,20]. TPO-RAs are initially approved to treat immune thrombocytopenia (ITP), aplastic anemia, and periprocedural thrombocytopenia [21]. Nevertheless, some adverse reactions are inevitable, such as headache, thrombosis and even increased bone marrow (BM) reticulin deposition [22,23]. Although several of these agents have been studied in treating CIT and are regarded as promising modalities [24,25], long-term clinical trials are still ongoing. Non-pharmacotherapies like platelet transfusions may cause transfusion-transmitted disease, infection, refractoriness and alloimmunization [26,27]. Additionally, high medical expenses caused by long-term administration incur a heavy financial burden and adversely affect the patient’s quality of life [28]. Hence, it is urgent and necessary to seek relatively inexpensive drugs with more effectiveness and fewer untoward events.

Traditional Chinese herbs provide a promising approach for the treatment of thrombocytopenia. As an important complementary and alternative therapy, medicinal herbs and their extracts have been widely used to prevent complications of CIT and radiation-induced thrombocytopenia (RIT), favoring medullary hematopoiesis and restoring immune function with relatively low costs and minor adverse reactions [29,30,31]. The dried caulis of *Polygonum multiflorum* Thunb is known as *Caulis Polygoni Multiflori* (CPM, also termed Shou-wu-teng in China). CPM is a widely applied medicinal herb and dietary ingredient with a long history [32]. CPM is recorded as a health food on the list of Chinese medical materials on the National Health Commission of the People’s Republic of China [33]. Pharmacological studies demonstrate that CPM has antioxidant, anti-obesity, anti-inflammatory, antibacterial, lowering blood glucose and immunomodulatory effects [34]. Studies have revealed its significant practical values that lie in alleviating cerebral stroke [35], sleep disorders [32], hypertension [36] and hyperglycemia [37]. It is well documented in the Chinese Pharmacopoeia that CPM can nourish blood and tranquilize the mind [38]. Furthermore, CPM is definitely recorded to treat anemia in an encyclopedia of Chinese medicinal substances named *Zhong Yao Daci Dian* [39]. Anemia exhibits impaired hemopoietic function, peripheral blood pancytopenia or even myelosuppression in severe cases [40]. It is worth noting that aplastic anemia is a prototype of reduced production, characterized as diminished erythrocytes and platelets as well [41]. The traditional application of CPM reveals its close contact with hematopoiesis. However, it is unclear whether CPM has therapeutic effects on thrombocytopenia.

In the present study, we first conducted the component analysis of CPM using a UHPLC Q–Exactive HF-X mass spectrometer (UHPLC QE HF-X MS). Then, we found that CPM was conducive to the MK differentiation of K562 and Meg-01 cells. Furthermore, the therapeutic effects of CPM on RIT were investigated in a whole-body-irradiated mice model. Finally, we explored the underlying mechanism of CPM action using RNA sequencing (RNA-seq) and experimental verification. Our findings demonstrate that CPM is a novel, promising thrombopoiesis-stimulating agent.

## 2. Results

### 2.1. Characterization of CPM

UHPLC QE HF-X MS was used to characterize the representative chemical constituents in the CPM. Based on the analysis of the molecular ions and fragmentation patterns, several representative components were characterized in the total ion chromatogram (Appendix A).

### 2.2. CPM induces MK Differentiation In Vitro

MK differentiation and maturation is one of the inevitable phases for platelet production, which is characterized by a large nucleus, polyploidization and increased expressions of megakaryocytic markers. Therefore, the effect of CPM on the MK differentiation of K562 and Meg-01 was detected. As a positive control, PMA (1 nM) induced the production of many large cells in K562 and Meg-01 cells after treatment for 5 days (Appendix A). Similarly, after treatment with CPM (20, 40 and 80 µg/mL) for 5 days, there were many large cells in the CPM-treated groups, while the control group had few large cells (Figure 1A). Giemsa staining further showed that these large cells had polylobulated nuclei in the CPM and PMA-treated groups, while the cells in the control group had a single nucleus (Figure 1B, Appendix A), which indicated that CPM contributed to increased size and multilobulated multinuclear. The expression of MK differentiation and maturation-associated antigens CD41 and CD42b was detected. As a result, PMA significantly increased the proportion of CD41^+^CD42b^+^ cells in the two cell lines (Appendix A). Similarly, CD41^+^CD42b^+^ cells were significantly increased after CPM treatment in the two cell lines in a concentration-dependent manner (Figure 1C–F). Furthermore, the ploidy assay showed notably increased number of 4 N and ≥8 N cells in the CPM and PMA-treated groups when compared with the control group in two cell lines (Figure 1G–J, Appendix A). These data suggest that CPM was able to promote MK maturation. Then, apoptosis was evaluated after CPM treatment. According to the results, the total apoptotic rate of the two cell lines all decreased after CPM treatment (Figure 2A–D). Furthermore, the cytotoxicity of CPM was measured by a lactate dehydrogenase (LDH) assay. The results demonstrated no significant difference in LDH release between the CPM-treated groups and the control group in the two cell lines (Figure 2E,F). In conclusion, these results suggest that CPM conspicuously promotes MK differentiation and maturation without cytotoxicity.

### 2.3. The Therapeutic Effects of CPM on Myelosuppressed Mice

Given that CPM could accelerate MK differentiation and maturation in vitro, the in vivo activities of CPM were further evaluated using an RIT mouse model. After administration with CPM (75, 150 and 300 mg/kg/d) and TPO (3000 U/kg) for 12 consecutive days, the circulating platelet levels in all irradiated mice reached nadir on day 7, with no remarkable difference among irradiated groups (Figure 3A). The number of platelets on days 10 and 12 was significantly higher in the groups treated with CPM and TPO in comparison with the model group (Figure 3A). Mean platelet volume (MPV) is related to platelet function and activation [42]. No difference was found among the groups, suggesting that CPM stimulated platelet generation without affecting its function and activation (Figure 3B). The numbers of white blood cells (WBCs) and red blood cells (RBCs) were also examined. We found that there were much more WBCs in the CPM (75 mg/kg) and TPO-treated groups compared with the control group on day 12, indicating that CPM could stimulate WBC recovery (Figure 3D). CPM (150 mg/kg and 300 mg/kg) treatment markedly improved RBC counts on days 10 and 12, while TPO had a similar effect on day 12 (Figure 3C). Collectively, CPM exerted a noteworthy effect on platelet recovery; in the meantime, it also contributed to WBC and RBC recovery. The animal body weight showed a marked increase in CPM (75 mg/kg and 150 mg/kg) and TPO-treated groups in comparison with the model group on day 12, suggesting that CPM protected the loss of body weight caused by irradiation (Figure 3E). The visceral indexes displayed that the indexes of the kidney and liver were significantly high in RIT mice, which recovered after CPM or TPO administration (Figure 3F,G). In addition, hematoxylin and eosin (H&E) staining suggested that there were no significant differences in the complete hepatic lobules and radial arrangement of central vein-centered hepatocytes between each group (Figure 3H). Compared with the control group, the kidney sections showed no significant pathological changes in the CPM-treated groups (Figure 3H). Together, these results demonstrate that CPM promotes platelet generation without a general adverse effect on mice.

### 2.4. CPM Promotes Megakaryopoiesis and Platelet Activation In Vivo

BM is generally thought to be the major region of thrombopoiesis from MKs [7]. To further explore the role of CPM in raising platelets in vivo, H&E staining was conducted to reveal whether CPM promoted megakaryopoiesis. The results demonstrated that the number of MKs in CPM (75, 150 and 300 mg/kg) and TPO-treated groups was higher than that of the model group (Figure 4A,B), indicating that CPM could enhance megakaryopoiesis in BM. Then, the surface antigens CD41 and CD61, known as MK differentiation markers, were measured in BM and spleen cells. The results revealed that the proportions of CD41^+^CD61^+^ cells in groups treated with CPM (75, 150 and 300 mg/kg) and TPO were obviously elevated when compared with the model group in BM and spleen cells (Figure 4C–E), indicating CPM promoted MK differentiation in BM and spleen. Moreover, we found a notable increase in proportions of 4 N and ≥8 N cells in BM and spleen after CPM (75, 150 and 300 mg/kg) and TPO treatments (Figure 4F–H), demonstrating that CPM administration stimulated MK maturation in BM and spleen. Correspondingly, CD41^+^CD61^+^ cell proportions in PB were markedly increased after CPM (75, 150 and 300 mg/kg) and TPO treatments (Figure 4I,K), suggesting that CPM accelerated MK differentiation in PB. Collectively, the above data suggested that CPM was able to accelerate platelet production by promoting MK differentiation and maturation in BM and spleen. Furthermore, p-selectin (also termed CD62p) was examined, which characterized platelet activation. As a result, the proportions of CD41^+^CD62p^+^ cells (platelets) in groups with CPM (75, 150 and 300 mg/kg) and TPO treatments were much higher than in the model group (Figure 4J,L). The percentage of CD41^−^CD62p^+^ cells (activated platelets) was significantly higher in the groups treated with CPM (150 and 300 mg/kg) and TPO than in the model group (Figure 4J,L). To sum up, all these results indicate that CPM enhances MK differentiation and maturation, platelet generation and activation without toxicity, which appears to have excellent therapeutic effects on RIT mice.

### 2.5. Gene Expression Profile in MK Differentiation Induced by CPM

The gene expression profile induced by CPM was detected using RNA-seq. Hierarchical clustering analysis revealed that the control and CPM-treated groups were clustered together, and there was a significant difference in gene expression between the two groups (Figure 5A). Volcano plots uncovered a total of 1950 differentially expressed genes (DEGs) between the control and CPM-treated groups, of which 850 were upregulated and 1100 were downregulated (Figure 5B, Appendix A). DO enrichment analysis revealed that CPM could treat bone disease, thrombocytopenia, blood platelet disease, bone marrow disease and blood coagulation disease (Figure 5C). GO enrichment results suggested that CPM could regulate positive regulation of cell differentiation, regulation of phosphatidylinositol 3-kinase signaling, positive regulation of hemopoiesis, MAPK cascade, myeloid cell differentiation, and positive regulation of MAP kinase activity, etc. (Figure 5D). KEGG pathway enrichment analysis revealed that the CPM regulated the MAPK signaling pathway, PI3K/Akt signaling pathway, hematopoietic cell lineage, Ras signaling pathway, as well as platelet activation (Figure 5E). Reactome pathway enrichment analysis further demonstrated that the DEGs were markedly enriched in platelet activation, signaling and aggregation, MAPK1/MAPK3 signaling, MAPK family signaling cascades, hemostasis, PI3K/Akt activation and platelet aggregation (Plug Formation) (Figure 5F). All these terms of GO, KEGG and Reactome enrichment analysis were closely related to MK differentiation, platelet production and platelet function. It was worth noting that PI3K/Akt and MAPK signaling pathways were the common terms in GO, KEGG and Reactome enrichment analysis, which suggested that PI3K/Akt and MAPK were crucial for CPM-induced MK differentiation.

### 2.6. CPM Induces MK Differentiation through PI3K/Akt and MEK/ERK (MAPK) Signaling Pathways

The expression of proteins belonged to PI3K/Akt and MEK/ERK signaling pathways was detected by western blotting. The results showed that CPM (20, 40 and 80 µg/mL) significantly induced the expression of IL-6 and IL1R1, located upstream of PI3K/Akt signaling pathway, activated the expression of p-PI3K and p-Akt (Figure 6A), which indicated that CPM activated PI3K/Akt signaling pathway. CPM (20, 40 and 80 µg/mL) could also significantly activate phosphorylation of RAS, MEK1/2 and ERK1/2 (Figure 6A), suggesting that CPM was able to activate MEK/ERK signaling pathway. Moreover, the expression of several transcription factors (TFs), including PBX1, GATA1, EGR1 and TAL1, involved in MK differentiation were measured. The results demonstrated that CPM (20, 40 and 80 µg/mL) obviously induced PBX1, GATA1, EGR1 and TAL1 expressions (Figure 6A). To further ascertain the close association of CPM action on MK differentiation with the above two signaling pathways, LY249002 and SCH772984, acknowledged inhibitors of PI3K/Akt and MEK/ERK signaling pathways were used, respectively. As expected, LY294002 and CPM co-treatment notably reduced the expression of CD41 and CD42b compared with CPM treatment (Figure 6B,C), which indicated that suppression of PI3K/Akt signaling pathway by LY294002 blocked MK differentiation induced by CPM. Similarly, combined treatment of SCH772984 and CPM attenuated CD41 and CD42b expression compared with CPM treatment (Figure 6D,E), suggesting that blockage of MEK/ERK signaling pathway by SCH772984 abolished CPM’s effect on MK differentiation. Based on the above results, we conclude that CPM activates PI3K/Akt and MEK/ERK signaling pathways, causing downstream activation of hematopoietic TFs (GATA1, TAL1, PBX1 and EGR1), thereby stimulating MK differentiation.

## 3. Discussion

Thrombocytopenia induced by radiotherapy or chemotherapy is a common cancer complication that leads to treatment delay, worse health issues, the risk of bleeding, and even death [43,44]. Currently, there are no effective agents for the treatment of RIT and CIT. Chinese herbal medicines have been an integral part of Chinese culture and medical practice. Many traditional Chinese medicines (TCMs), whether in the single herb or formula form, have been well-reported to be effective in treating thrombocytopenia, such as an herbal decoction of *Radix angelicae sinensis* and *Radix astragali* (Danggui Buxue Tang, DBT) [29] and ginseng [45] with affordable price and easy access.

Through activity screening plenty of TCMs in vitro, we identified that CPM was able to increase the cell size, enhance nucleoli number, promote CD41 and CD42b expression, and increase polyploidization of K562 and Meg-01 cells, demonstrating that CPM could promote MK differentiation and maturation. As is well known, K562 and Meg-01 cells are classical models for exploring MK differentiation. However, as human erythroleukemic cell lines and human megakaryoblastic leukemia cell lines, their ability to differentiate into MK is limited [46,47]. This may be a reason why CPM has modest effects on the MK differentiation of K562 and Meg-01 cells. In future work, we will establish human CD34^+^-derived MKs system, which can reproduce the events of normal megakaryopoiesis clearly [48]. As the precursor of platelets, increased MK differentiation and maturation may be conductive to platelet production. Therefore, the in vivo activity of CPM was assessed using an RIT mouse model. Our results demonstrated that CPM could significantly enhance platelet and RBC recovery after irradiation injury. Interestingly, it has been clearly documented in the Chinese Pharmacopoeia that CPM can nourish blood and tranquilize the mind [38], which represents a close connection with blood. In addition, it is definitely recorded that CPM is used to treat anemia in *Zhong Yao Daci Dian* [39]. Anemia, equivalent to blood deficiency syndrome, appears to diminish blood cells [40]. Our findings are in accordance with the traditional use of CPM in blood diseases. In consideration of clinical translation, the toxicity of CPM was evaluated. The results showed that CPM could rescue hepatomegaly and renomegaly induced by irradiation. In addition, CPM did not cause any pathological changes in the organ structures of the liver and kidney. These findings suggest that CPM has potential for clinical application. The increased platelet level may be caused by enhanced megakaryopoiesis. We found that CPM treatment contributed to a greater size and MK count in the BM, indicating that CPM accelerated megakaryopoiesis in the BM. Flow cytometry further revealed that CPM had the ability to promote MK differentiation and maturation in BM, which is in line with the in vitro results. Spleen, a major extramedullary hematopoiesis organ, is of pivotal importance in platelet biogenesis [49]. We found that CPM administration could also increase MK number, differentiation and maturation in the spleen, indicating that CPM stimulated extramedullary hematopoiesis. In general, all these results suggest that both enhanced megakaryopoiesis, MK differentiation and maturation in BM and spleen contribute to increased platelet levels induced by CPM.

RNA-seq was applied to elucidate the molecular mechanism of CPM in promoting MK differentiation. The multiple analysis of RNA-seq data collectively points to the common signaling pathways: PI3K/Akt and MAPK signaling pathways, which are closely related to CPM-induced MK differentiation. PI3K/Akt and MAPK signaling pathways deliver intracellular signal cascades and have been implicated in the modulation of multiple cellular processes, such as cell proliferation, differentiation and apoptosis [50,51]. Moreover, aberrations in PI3K/Akt and MAPK signaling pathways can cause a broad range of diseases, such as metabolic diseases, neurodegenerative diseases, blood diseases and various cancers [52,53,54,55]. The MAPK pathway is composed of three main signaling cassettes: MEK/ERK, p38 and JNK [55]. It is known that PI3K/Akt and MAPK signaling pathways positively regulate megakaryopoiesis and thrombopoiesis. Activation of the MAPK pathway is necessary for the development and maturation of MK progenitors (MkPs), while AKT regulates the cell cycle of MkPs by downregulating the Forkhead box O (FOXO) TF family [56]. For example, ingenol has been validated to induce MK differentiation by activating PI3K/Akt signaling pathway [57]. Similarly, ERK1/2 and JNK (but not P38) MAPK pathways have been demonstrated to be involved in megakaryopoiesis [58]. It is explicit that pleiotropic cytokines upstream of these pathways trigger the intracellular signaling involved in megakaryopoiesis and thrombocytopoiesis through their receptors, among which, IL6 and IL1β are two representative pleiotropic growth factors besides TPO [59]. IL6 has been proven to be a potent stimulator in the maturation of megakaryocytic lineages in vitro and in vivo, which can effectually prevent thrombocytopenia [10,60]. Moreover, IL6 is capable of correcting anemia in combination with IL3, making rapid activation of platelets by thrombin and platelet activating factor, and exhibiting a procoagulant effect on ameliorating bleeding propensity [60,61]. IL1β gives rise to the activation of PI3K/Akt and MAPK (ERK and p38) pathways through binding to its receptor, IL1R1, ultimately resulting in an increase in MK maturation. Additionally, in conjugation with IL1R1 on platelet, ILlβ enhances hemostasis function of platelet [62].

In accordance with RNA-seq results, western blot analysis demonstrated that CPM significantly increased the expression of proteins belonged to PI3K/Akt (IL6, IL1R1, p-PI3K, p-Akt) and MAPK (RAS, p-MEK, p-ERK) signaling pathways. Moreover, CPM markedly promoted the expression of several hematopoietic TFs (GATA1, TAL1, PBX1, EGR1). These TFs play a fundamental role in MK development and thrombocytopoiesis. GATA1 is required for MK differentiation, maturation and platelet formation. GATA1 facilitates cell growth and DNA content in MKs by promoting cyclin D1 expression, leading to appropriate modulation of the cell cycle and differentiation [63,64]. TAL1 controls the coordination of MkP proliferation and differentiation, which are required for megakaryopoiesis. Importantly, TAL1 is advantageous for the completion of polyploidization and terminal maturation in MkPs by mediating the transcriptional control of p21, a cyclin-dependent kinase (CDK) inhibitor [65]. PBX1, a proto-oncogene, is necessary for hematopoietic development. A lack of PBX1 leads to a diminished number and impaired functions of megakaryocyte-erythroid progenitors (MEPs). Furthermore, PBX1 has been reported to support the expansion of MkPs and improve MK differentiation [66,67]. EGR1 can regulate cell survival, growth, apoptosis and differentiation, which shows a crucial effect on MK differentiation [68,69]. In sum, these TFs play a key role in divergent aspects of MK development.

To verify whether CPM-induced MK differentiation was through activating PI3K/Akt and MEK/ERK signaling pathways, pharmacological blocking of those pathways was applied using their inhibitors, SCH772984 and LY294002, respectively. As expected, LY294002 and SCH772984 almost completely eliminated the MK differentiation induced by CPM, which demonstrated that CPM contributed to MK differentiation via activating PI3K/Akt and MEK/ERK signaling pathways. Collectively, we conclude that CPM can increase IL6 and IL1R1 expressions, followed by triggering intracellular PI3K/Akt and MEK/ERK signaling pathways, subsequently upregulating PBX1, GATA1, EGR1 and TAL1, ultimately leading to MK differentiation and platelet generation (Figure 7).

Since CPM has multi-components, it is reasonable that different components may regulate multi-targets and multi-pathways. Except for 11 components identified in this study, there must be other components in CPM and there might be some interacting effects of them. It has been reported that nobiletin can activate MAPK/ERK signaling pathway, subsequently activating EGR1 expression, ultimately leading to MK differentiation of K562 cells [69]. Therefore, nobiletin may be the main active compound of CPM, which contributes to the activation of MEK/ERK signaling pathway and EGR1 expression. A study has shown that nodakenin is able to promote adult hippocampal neurogenesis and improve cognitive function by activating Akt phosphorylation [70]. It is indicated that nodakenin may induce PI3K/Akt signaling pathway of K562 cells. Chrysophanol, an anthraquinone, exhibits an anti-proliferation effect on choriocarcinoma cells JEG-3 via activating PI3K/Akt and ERK signaling pathways [71]. It is possible that a similar mechanism of action of chrysophanol may appear in the MK differentiation of K562 cells. Fisetin, a flavonoid, suppresses proliferation and metastasis via activation of MEK/ERK signaling pathway [72]. Fisetin can also block high glucose (HG)-induced neurotoxicity in HT22 cells through stimulating PI3K/Akt signaling pathway [73]. Moreover, maslinic acid has been reported to improve cognitive function in the scopolamine-induced memory impairment mouse model by activating PI3K/Akt and ERK/CREB signaling pathways [74]. Hence, fisetin or maslinic acid may be the compounds of CPM that regulate PI3K/Akt or MEK/ERK signaling pathways in K562 cells. The relationship between these compounds of CPM and their mechanisms of action are complicated. In future work, we will identify the active compounds of CPM, elucidate their molecular mechanisms in regulating MK differentiation and platelet formation, and provide more potential agents for the treatment of thrombocytopenia.

## 4. Materials and Methods

### 4.1. Chemicals

CPM was purchased from Sichuan New Green Pharmaceutical Science and Technology Development Co., Ltd. (Chengdu, China). An appropriate amount of CPM was weighed and solubilized in sterilized deionized water: dimethyl sulfoxide (1:1, *v*/*v*) for cell treatment, while animals were intragastrically administrated with CPM dissolved in normal saline.

### 4.2. Sample Preparation

Fifty milligrams of CPM was solubilized in 400 µL methanol: water (4:1, *v*/*v*) solution with 0.02 mg/mL L-2-chlorophenylalanin as internal standard. The mixture was allowed to settle at –10 °C and treated with a high throughput tissue crusher Wonbio-96c (Shanghai wonbio biotechnology Co., Ltd.) at 50 Hz for 6 min, then put in an ultrasonic bath (frequency: 40 kHz) for 30 min at 5 °C. The samples were placed at −20 °C for 30 min to precipitate proteins followed by centrifugation at 13,000× *g*/min for 15 min at 4 °C. Finally, the supernatant was carefully transferred to sample vials for LC-MS/MS analysis.

### 4.3. UPLC-MS Conditions

The UHPLC Q–Exactive HF-X mass spectrometer (UHPLC QE HF-X MS) system of Thermo Fisher Scientific equipped with an electrospray ionization (ESI) source operating in positive ion mode was used to analyze CPM. The separation was performed on ACQUITY UPLC HSS T3 (100 mm × 2.1 mm, 1.8 µm). The mobile phase consisted of 0.1% formic acid in water: acetonitrile (95:5, *v*/*v*) (solvent A) and 0.1% formic acid in acetonitrile: isopropanol: water (47.5:47.5:5, *v*/*v*) (solvent B). The linear gradient conditions were as follows: from 0 to 0.1 min, 0–5% B; from 0.1 to 2 min, 5–25% B; from 2 to 9 min, 25–100% B; from 9 to 13 min, 100–100% B; from 13 to 13.1 min, 100–0% B; and from 13.1 to 16 min, 0–0% B for equilibrating the systems. The flow rate was 0.30 mL/min. The column temperature was 45 °C.

The conditions of MS analysis were as follows: heater temperature, 400 °C; Capillary temperature, 320 °C; sheath gas flow rate, 40 arb; Aux gas flow rate, 10 arb; ion-spray voltage floating (ISVF), −2800 V in negative mode and 3500 V in positive mode, respectively; normalized collision energy, 20–40–60 V rolling for MS/MS. Full MS resolution was 70,000, and MS/MS resolution was 17,500. Data acquisition was performed with the Data Dependent Acquisition (DDA) mode and the mass range of the detection was 70–1050 m/z.

### 4.4. Cell Culture

The human chronic myeloid leukemia (CML) cell line K562 and the human megakaryoblastic leukemia cell line Meg-01 were purchased from the American Type Culture Collection (Bethesda, MD, USA). RPMI-1640 medium (Gibico, Invitrogen Corporation, Carlsbad, CA, USA) supplemented with 10% (*v*:*v*) fetal bovine serum (FBS, Thermo Fisher Scientific, Waltham, MA, USA) and 1% (*v*:*v*) penicillin and streptomycin were utilized for cell cultivation at 37 °C in a humidified atmosphere of 5% CO_2_.

### 4.5. LDH Assay

The LDH assay was performed using an LDH assay kit (LEAGENE, Beijing, China) in accordance with the manufacture’ protocol. Briefly, the K562 and Meg-01 cells were planted in 96-well plates (5 × 10^3^ per well). Maximum LDH control was used to assess the total amount of LDH present in the cells. The cells were incubated for the indicated days. Lysis solution was added to the maximum LDH control to induce the LDH release of cells. After the maximum LDH control was treated with lysis solution for 1 h, a series of LDH reagents were added to the supernatant of each group to conduct the enzyme catalytic reaction. Finally, the absorbance was measured at 440 nm.

### 4.6. Cell Apoptosis Assay

An Annexin V-FITC/PI Apoptosis Detection kit (BD Biosciences, San Jose, CA, USA) was used to determine the apoptotic effect of CPM according to the manufacturer’s instructions. Cells were incubated in 12-well plates at a density of 2 × 10^4^ cells/mL with or without CPM (20, 40 and 80 µg/mL) treatment for 5 days. Following treatment, the cells were harvested, washed twice with cold phosphate buffer saline (PBS) and resuspended in 100 μL binding buffer. Next, Annexin V-FITC and propidium iodide (PI) were added to the cell suspension and incubated at room temperature for 15 min in the dark. Cells were immediately analyzed by BD FACSCanto II flow cytometer (BD Biosciences, San Jose, CA, USA) and analysis was done by FlowJo software. Untreated cells served as the negative control.

### 4.7. Cell Morphological Observation

Cells (2 × 10^4^) were seeded in 12-well plates at a density of 2 × 10^4^ cells/mL. After treatment with CPM (20, 40 and 80 µg/mL) or positive control PMA (1 nM) for 5 days, the change in cell morphology was randomly observed by microscopy (NIKON, Japan) at 10× resolution.

### 4.8. Giemsa Staining

The cells with or without drug treatment were harvested from 12-well plates on the fifth day and washed three times with ice-cold PBS. After fixing with methanol and glacial acetic acid mixture (3:1) for 5 min, the cells were placed onto glass slides and stained with Giemsa Working solution consisting of 1 part of Giemsa Stock Solution and 9 part of Giemsa Buffer (Sigma, St. Louis, MO, UAS) for 10 min at room temperature. The cells were washed with distilled water and viewed under a microscope (NIKON, Tokyo, Japan).

### 4.9. Surface Marker Analysis

Flow cytometry was performed to analyze the expression of CD41 and CD42b in K562 and Meg-01 cells. Briefly, the cells were seeded at a density of 2 × 10^4^ cells/mL in 12-well plates and incubated with CPM (20, 40 and 80 µg/mL) or positive control PMA (1 nM) for 5 days. Cells were harvested, resuspended in 100 µL PBS and stained with FITC-conjugated anti-CD41 and PE-conjugated anti-CD42b (Biolegend, San Diego, CA, USA) for 30 min on ice in the dark. The expression of CD41 and CD42b was detected by BD FACSCanto II flow cytometer (BD Biosciences, San Jose, CA, USA) and the results were analyzed by FlowJo software.

### 4.10. Polyploidy Analysis

Cells were collected, washed with PBS twice and permeabilized with ice-cold 70% ethanol at 4 °C till use. According to the manufacturer’s protocol, the cells were washed twice and then treated with a trypsin inhibitor/RNase buffer using the CycleTEST™ PLUS DNA Reagent Kit (BD Biosciences, San Jose, CA, USA). After that, the cells were incubated with a PI stain solution for 10 min. The ploidies of the samples were detected by a BD FACSCanto II flow cytometer (BD Biosciences, San Jose, CA, USA) and analyzed by FlowJo software.

### 4.11. Animals

All animal experiments were conducted according to the National Guidelines for Experimental Animal Welfare and approved by the Ethics Committee of Southwest Medical University (License No. 20211123-014). Kunming (KM) mice aged 6–8 weeks and weighing 18–22 g were purchased from Tengxin Biotechnology Co., Ltd. (license No. SCXK (Beijing, China) 2019–0010; Chongqing, China). The mice were housed in a controlled facility with an ambient temperature of (24 ± 2) °C and a relative humidity range from 45% to 50% under a 12-h light-dark cycle, and supported with a standard diet and water ad libitum. The mice were randomly assigned to six groups: control, model (X-ray irradiation), TPO (3000 U/kg), and different doses of CPM (75, 150 and 300 mg/kg) groups. Except for the control, all the mice underwent 4 Gy X-ray total-body irradiation after acclimating for 7 days. The mice in the control and model groups were treated intragastrically with normal saline (0.1 mL per 10 g weight). The others were administered TPO intraperitoneally or in distinct doses of CPM intragastrically.

### 4.12. Routine Blood Test

During CPM administration, PB (40 µL) was harvested from the eyes’ venous plexus and treated with 160 µL diluent. The samples were subjected to peripheral hemogram analysis using a full-automatic blood cell analyzer (SYSMEX XT-1800Iv; Kobe, Japan).

### 4.13. Visceral Index

On the indicated day, the mice were sacrificed by cervical dislocation. The liver and kidney of each group were separated, washed with normal saline solution, dried with filter paper and finally weighed right away to calculate visceral index by the following equation: visceral index (mg·g^−1^) = visceral organ wet weight (mg)/body weight (g) × 100%.

### 4.14. Histology Analysis

After 12 days of administration of CPM, the tibias, kidney and liver were separated from 3 mice randomly-picked from each group. All organs were fixed with 10% paraformaldehyde for 24 h. The tibias were decalcified with a decalcifying solution for more than one month. Then, all the organs were embedded in paraffin and cut into 5 µm thick sections followed by deparaffinization and stained with H&E. The samples were observed under an Olympus BX51 microscope (Olympus Optical, Tokyo, Japan). Visual fields of each sample were randomly shot and the number of MKs was counted.

### 4.15. Flow Cytometry Analysis of PB Cells

Fifty microliters of PB was collected from the venous plexus and immediately transferred into 2 mL EP tubes preloaded with 1 mL citrate solution and 130 µL flow cytometric preservation solution. According to the manufacturer’s instructions, 0.5 µL FITC conjugated anti-CD41, 1.25 µL PE conjugated anti-CD61, 0.5 µL FITC conjugated anti-CD41 and 1.25 µL PE conjugated anti-CD62p were added to the blood and incubated for 15 min at room temperature, respectively. After that, the samples were resuspended in 400 µL PBS for analysis using a BD FACSCanto II flow cytometer (BD Biosciences, San Jose, CA, USA).

### 4.16. Flow Cytometry Analysis of BM and Spleen Cells

Mice BM cells were flushed from the thighbone with 2 mL PBS. Spleen was grounded into single cells filtered by a 200 mesh nylon net. RBC lysis solution was used to remove RBCs from the whole harvested BM and spleen cells on ice. After being washed with PBS twice, cells (1 × 10^6^) were collected in 100 µL PBS and labeled with FITC conjugated anti-CD41 and PE conjugated anti-CD61 for 15 min on ice. For polyploidy analysis of the CD41^+^ cells, cells from the BM and spleens were treated with CD41 monoclonal antibody for 15 min and then incubated with PI stain solution for 15 min at room temperature. The polyploidy was analyzed using a BD FACSCanto II flow cytometer (BD Biosciences, San Jose, CA, USA).

### 4.17. RNA-Seq and Data Analysis

The K562 cells were treated with CPM (80 µg/mL) for 3 days. Total RNA of cells was extracted using TRIzol^®^ Reagent (Invitrogen, Carlsbad, CA, USA) based on the manufacturer’s instructions. RNA integrity was detected with a 2100 Bioanalyzer (Agilent Technologies, Santa Clara, CA, USA). RNA purity was assessed using a NanoDrop ND-2000 Spectrophotometer (Thermo Scientific, ND-2000, Waltham, MA, USA). One microgram of total RNA was used to construct an RNA-sequence (RNA-seq) transcriptome library using the TruSeqTM RNA sample preparation Kit from Illumina (San Diego, CA, USA). After that, RNA-seq was performed using the Illumina HiSeq xten/NovaSeq 6000 sequencer (2 × 150-bp read length) by Shanghai Majorbio Bio-pharm Biotechnology Co., Ltd. (Shanghai, China). The raw paired-end reads were processed with SeqPrep (https://github.com/jstjohn/SeqPrep) (accessed on 3 March 20022) and Sickle (https://github.com/najoshi/sickle) (accessed on 3 March 2022). Then, clean reads were aligned to the reference genome using HISAT2 (http://ccb.jhu.edu/software/hisat2/index.shtml) (accessed on 3 March 2022) software [75]. Transcripts were assembled with StringTie (https://ccb.jhu.edu/software/stringtie/index.shtml?t=example) (accessed on 3 March 2022) [76]. All the sequencing data are available in NCBI Sequence Read Archive (SRA) under the accession number GSE212290.

### 4.18. Differential Expression Analysis and Functional Enrichment

The gene expression level was assessed using the transcript per million reads (TPM) method. Gene abundances were quantified using RSEM software (http://deweylab.biostat.wisc.edu/rsem/) [77]. EdgeR was used for differential expression analysis [78]. The genes with a *p* value < 0.05 and log2FC > 1.5 were regarded as DEGs between the control and CPM-treated groups. Disease Ontology (DO) enrichment analysis, Gene Ontology (GO) enrichment analysis, Kyoto Encyclopedia of Genes and Genomes (KEGG) pathway analysis, Reactome pathway enrichment analysis and were performed by Disease Ontology database (http://disease-ontology.org) (accessed on 13 March 2022) [79], Goatools (https://github.com/tanghaibao/Goatools) (accessed on 13 March 2022), KOBAS (http://kobas.cbi.pku.edu.cn/home.do) (accessed on 13 March 2022), and Reactome Database (http://reactome.org/) (accessed on 13 March 2022), respectively.

### 4.19. Western Blotting

K562 cells treated with or without CPM (20, 40 and 80 µg/mL) were harvested after different treatments for 5 days. The cells were lysed with ice-cold RIPA lysis buffer (CST, MA, UAS) containing protease inhibitors (Sigma, St Louis, MO, USA). The protein was measured by the Quick Start™ Bradford 1× Dye Protein Assay Reagent (Bio-Rad, Hercules, CA, USA). Equal amounts of protein were loaded onto sodium dodecyl sulfate polyacrylamide gel electrophoresis (SDS-PAGE) gels and then electrotransferred onto polyvinylidene fluoride (PVDF) membranes. The membranes were blocked with 10% (*w*/*v*) skimmed milk for 1 h and then incubated with primary antibodies overnight at 4 °C followed by horseradish peroxidase (HRP)-conjugated secondary antibodies (Cell Signaling Technology, Danvers, MA, USA). Finally, visualization was performed using the ChemiDoc MP Imaging System (Bio-Rad, Hercules, CA, USA) after detection with an ECL Western Blotting detection reagent (4A Biotech Co., Ltd., Beijing, China). The protein bands were quantified using ImageJ software (NIH, Bethesda, MD, USA). The PVDF membranes were probed with the following antibodies: IL6 (Proteintech, 21865-1-AP), IL1R1 (Abmart, PK06658S), RAS (Abmart, T56672S), p-ERK (CST, 4370S), ERK (CST, 4696S), p-MEK (CST, 9154T), MEK (CST, 4694S), p-PI3K (Abmart, TA3241), PI3K (Abmart, TA5112), p-AKT (Abmart, TA0016S), AKT (Abmart, TU421951S), TAL1 (Proteintech, 55317-1-AP), EGR1 (Abmart, T57177), GATA1 (CST, 3535S), PBX1 (Proteintech, 18204-1-AP).

### 4.20. Statistical Analysis

All the results in this study were presented as mean ± standard deviations (SDs) using GraphPad Prism 7.0 software. Experiments were conducted in triplicate, unless stated otherwise. Statistical significance among groups was performed by analysis of variance (ANOVA), followed by Tukey–Kramer post-hoc analysis. A *p* value < 0.05 was considered as a statistical difference.

## 5. Conclusions

In summary, our study for the first time demonstrates that CPM exerts an active effect on MK differentiation and platelet production, which shows an excellent ability for the treatment of RIT. This study provides the modern medicine interpretation of CPM for its traditional use and provides a potential thrombopoietic agent for the treatment of thrombocytopenia.

## Figures and Tables

**Figure 1 pharmaceuticals-15-01204-f001:**
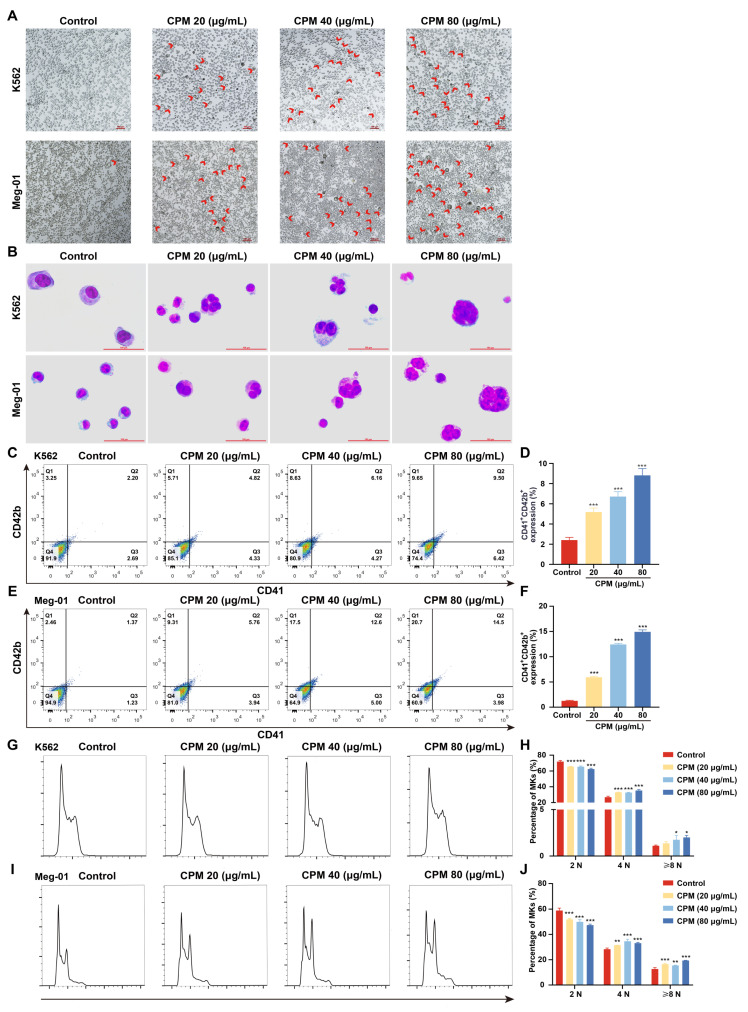
CPM induces MK differentiation. (**A**) Microscope photographs of K562 and Meg-01 cells with or without CPM (20, 40 and 80 µg/mL) treatment for 5 days. Scar bar: 100 µm. Microscopy fields were captured randomly at 10× resolution. (**B**) Giemsa staining of K562 and Meg-01 cells with or without CPM (20, 40 and 80 µg/mL) treatment for 5 days. Scar bar: 100 µm. (**C**,**E**) CD41 and CD42b expression of K562 and Meg-01 cells with or without CPM (20, 40 and 80 µg/mL) treatment for 5 days. (**D**,**F**) The proportion of CD41^+^CD42b^+^ cells in control and CPM-treated groups. Data are mean ± SD (n = 3). (**G**,**I**) DNA ploidy analysis of K562 and Meg-01 cells with or without CPM (20, 40 and 80 µg/mL) treatment for 5 days. (**H**,**J**) The percentage of 2 N, 4 N and ≥ 8 N cells in each group. Data are mean ± SD (n = 3, ANOVA). * *p* < 0.05, ** *p* < 0.01, *** *p* < 0.001 vs. the control group.

**Figure 2 pharmaceuticals-15-01204-f002:**
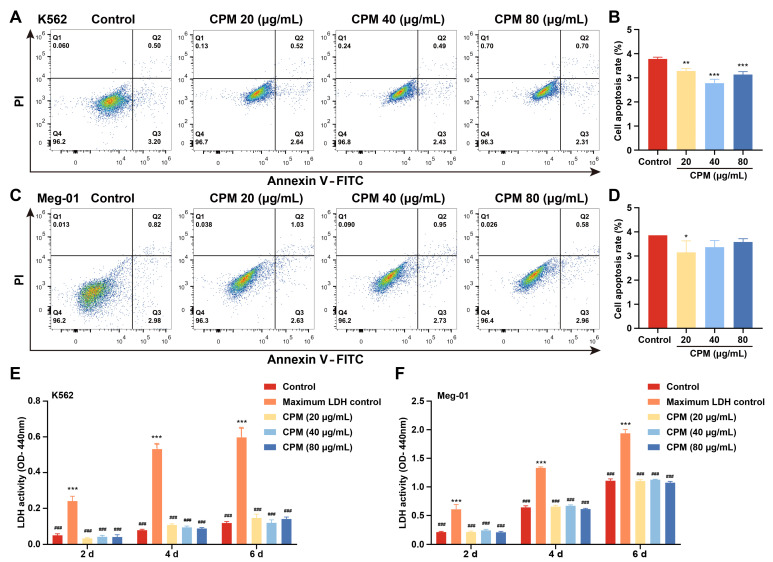
Cytotoxicity of CPM. (**A**,**C**) Apoptosis analysis in K562 and Meg-01 cells after treatment with or without CPM (20, 40 and 80 µg/mL) for 5 days. Q2 (Annexin^+^ PI^+^) denotes late apoptotic. Q3 (Annexin^+^ PI^−^) denotes early apoptotic cells. (**B**,**D**) Q2 + Q3 represents total apoptotic cells. Statistical analysis of apoptotic cells (Q2 + Q3) of K562 and Meg-01 cells in each group. Data are means ± SD (n = 3). (**E**,**F**) Detection of LDH activity of K562 and Meg-01 cells after treatment with or without CPM (20, 40 and 80 µg/mL) for 2, 4 and 6 days. Maximum LDH control represents the total amount of LDH present in the cells. Data are mean ± SD (n = 3, ANOVA). * *p* < 0.05, ** *p* < 0.01, *** *p* < 0.001 vs. the control group. ^###^ *p* < 0.001 vs. the maximum LDH control.

**Figure 3 pharmaceuticals-15-01204-f003:**
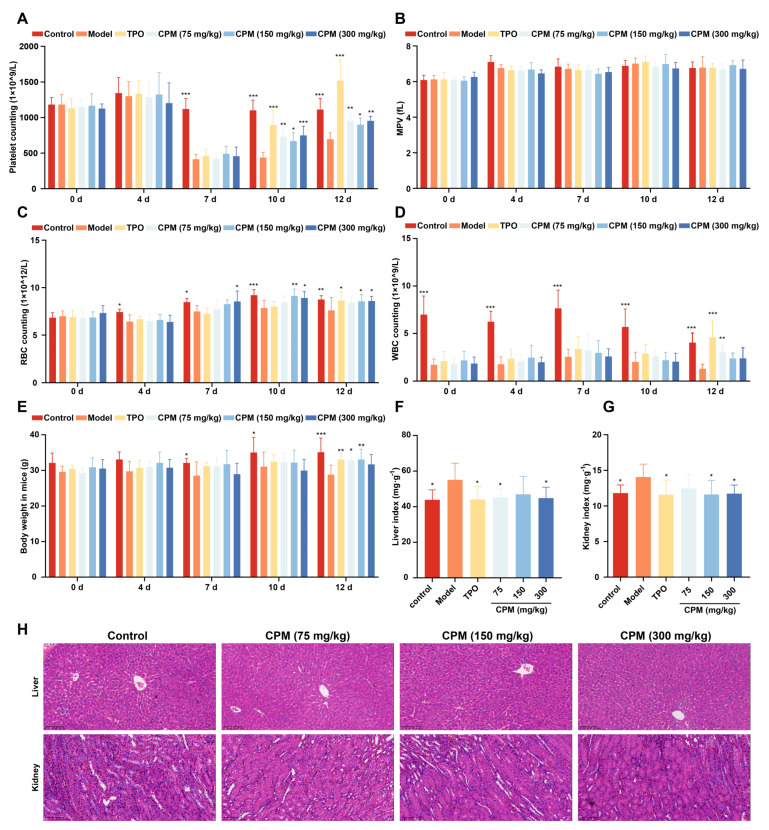
The therapeutic effects of CPM on RIT mice. The KM mice aged 6–8 weeks were irradiated and then administrated with normal saline, TPO (3000 U/kg), or CPM (75, 150 and 300 mg/kg) solubilized in normal saline for 12 days, respectively. Each group included 8 randomly assigned mice (4 male mice and 4 female mice). Routine blood examination was conducted on days 0, 4, 7, 10 and 12. (**A**) Platelet counts in each group. Data are mean ± SD (n = 8). (**B**) MPV in each group. Data are mean ± SD (n = 8). (**C**) RBC counts in each group. Data are mean ± SD (n = 8). (**D**) WBC counts in each group. Data are mean ± SD (n = 8). (**E**) Body weight in each group. Data are mean ± SD (n = 8). (**F**) Liver index in each group on day 12. Data are mean ± SD (n = 8). (**G**) Kidney index in each group on day 12. Data are mean ± SD (n = 8, ANOVA). * *p* < 0.05, ** *p* < 0.01, *** *p* < 0.001 vs. the model group. (**H**) H&E staining of the liver and kidney in each group on day 12. Bars: 200 µm.

**Figure 4 pharmaceuticals-15-01204-f004:**
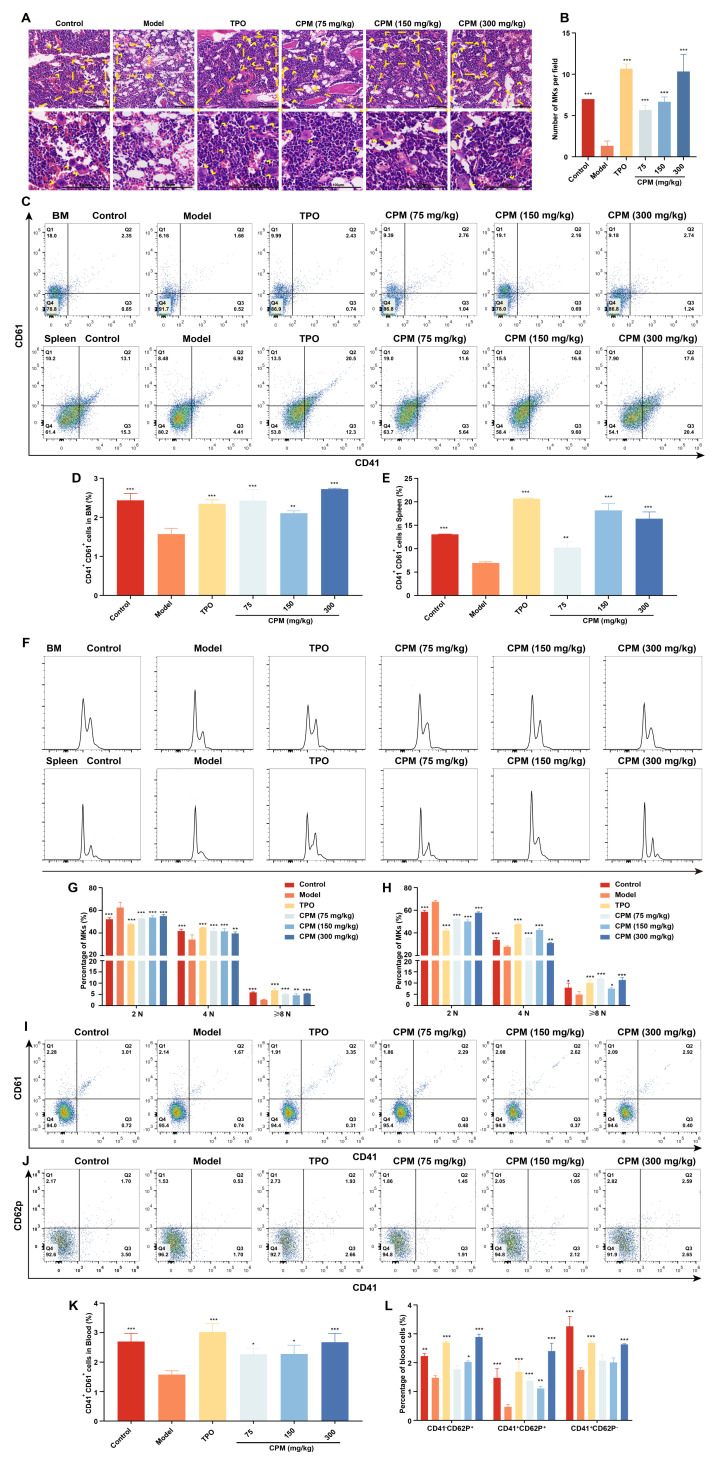
CPM promotes MK differentiation, maturation and platelet activation in RIT mice. (**A**) H&E staining of BM in control, model, TPO and CPM-treated groups on day 10. Bars: 100 µm. The MKs are indicated by arrows. (**B**) The MK counts of each group in BM. Data are mean ± SD (n = 3). (**C**) CD41 and CD61 expression in BM and spleen cells of each group on day 12. (**D**,**E**) The percentage of CD41^+^CD61^+^ cells in BM and spleen, respectively. Data are mean ± SD (n = 3). (**F**) Ploidy analysis of the BM and spleen cells. (**G**,**H**) The percentage of 2 N, 4 N and ≥ 8 N cells of BM and spleen in each group. Data are mean ± SD (n = 3). (**I**) The expression of CD41 and CD61 in PB of each group after treatment for 12 days. (**J**) The expression of CD41 and CD62p in PB of each group after administration for 12 days. (**K**) The proportion of CD41^+^CD61^+^ cells of PB in each group. Data are mean ± SD (n = 3). (**L**) The proportion of CD41^−^CD62P^+^, CD41^+^CD62P^+^ and CD41^+^CD62P^−^ cells in each group. Data are mean ± SD (n = 3, ANOVA). * *p* < 0.05, ***p* < 0.01, *** *p* < 0.001 vs. the model group.

**Figure 5 pharmaceuticals-15-01204-f005:**
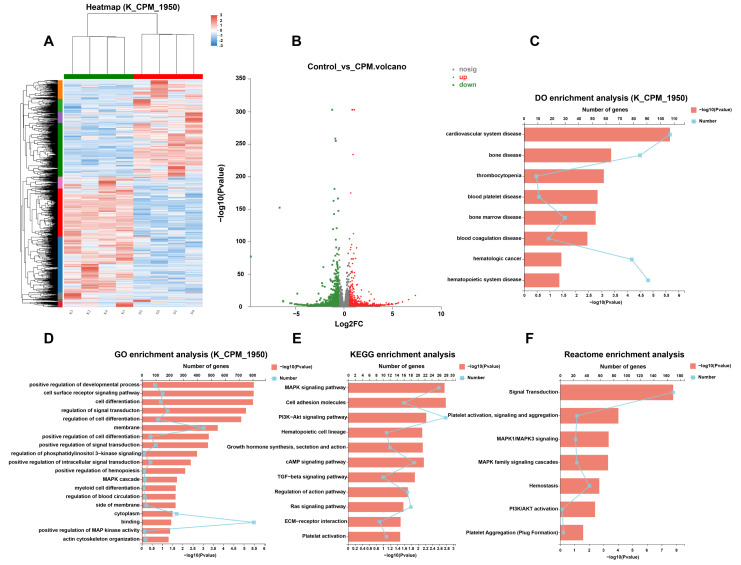
RNA-seq analysis of the gene expression profile modulated by CPM. (**A**) Hierarchical clustering analysis indicates the change in gene expression between the control and CPM-treated groups. (**B**) Volcano plot representing the DEGs regulated by CPM. (**C**) DO enrichment analysis. (**D**) GO enrichment analysis. (**E**) KEGG pathway enrichment analysis. (**F**) Reactome pathway enrichment analysis.

**Figure 6 pharmaceuticals-15-01204-f006:**
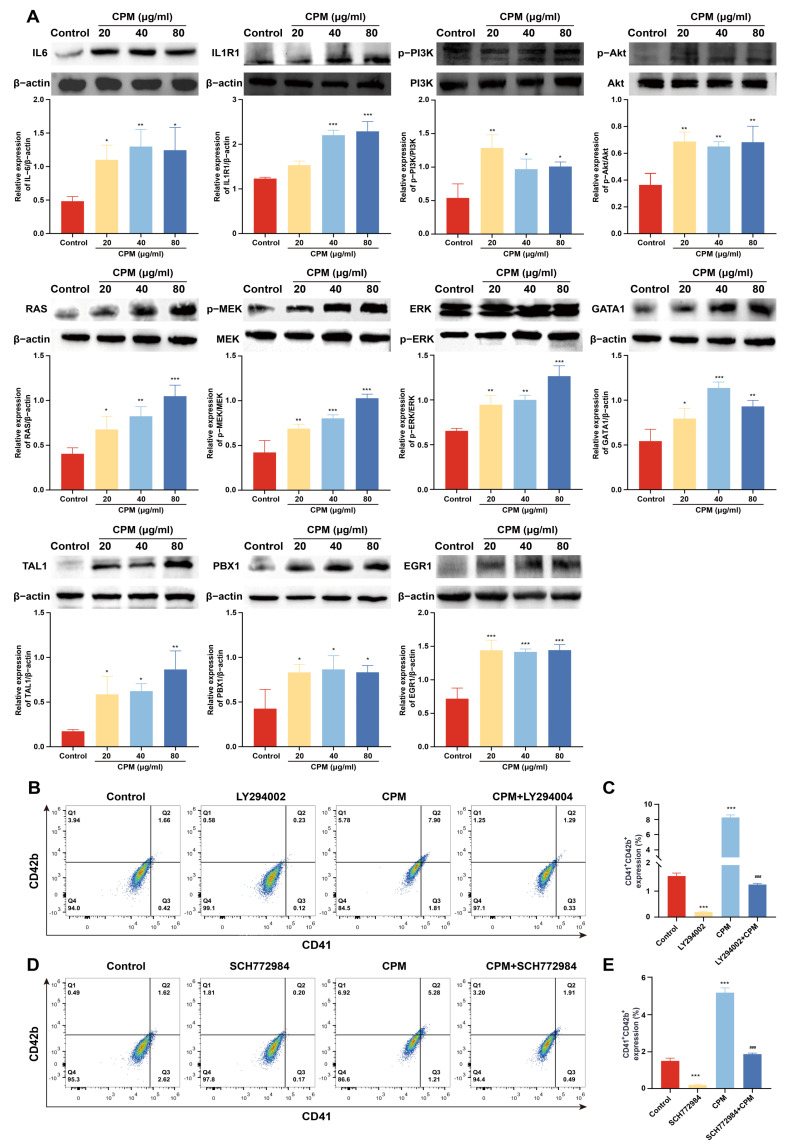
CPM induces MK differentiation through activating PI3K/Akt and MEK/ERK signaling pathways. (**A**) The measurement of proteins belonging to PI3K/Akt and MEK/ERK signaling pathways and hematopoietic TFs by western blot in control and CPM (20, 40 and 80 µg/mL)-treated groups on K562 cells. The data are the mean ± SD (n = 3). * *p* <0.05, ** *p* <0.01, *** *p* < 0.001 vs. the control group. (**B**) Detection of CD41 and CD42b expression in K562 cells after treatment with CPM (80 µg/mL), CPM (80 µg/mL) + LY294002 (20 µM), and LY294002 (20 µM) for 5 days. (**C**) The proportion of CD41^+^CD42b^+^ cells in each group. Data are mean ± SD (n = 3). *** *p* < 0.001 vs. the control group. ^###^ *p* < 0.001 vs. CPM group. (**D**) Detection of CD41 and CD42b expression on K562 cells after CPM (80 µg/mL), CPM (80 µg/mL) + SCH772984 (3 µM), and SCH772984 (3 µM) treatments for 5 days. (**E**) The percentage of CD41^+^CD42b^+^ cells in each group. Data are mean ± SD (n = 3, ANOVA). *** *p* < 0.001 vs. the control group. ^###^ *p* < 0.001 vs. the CPM group.

**Figure 7 pharmaceuticals-15-01204-f007:**
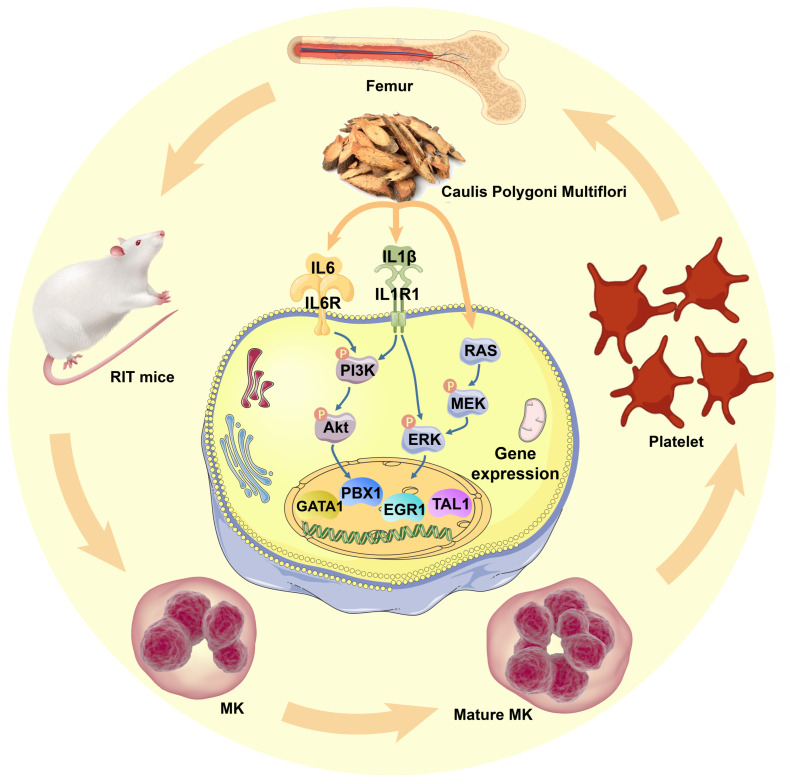
Schematic model for CPM action in regulating megakaryopoiesis and thrombopoiesis. CPM induces IL-6 and IL1R1 expressions, causing the activation of PI3K/Akt and MEK/ERK (MAPK) signaling pathways, followed by the activation of downstream hematopoietic TFs, including GATA1, TAL1, PBX1 and EGR1, which stimulate MK differentiation and platelet production.

## Data Availability

Data is contained within the article and Appendix A.

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
