# Peer review of "Caulis Polygoni Multiflori Accelerates Megakaryopoiesis and Thrombopoiesis via Activating PI3K/Akt and MEK/ERK Signaling Pathways"

_pharmaceuticals, 2022, doi:10.3390/ph15101204_

Round 1

Reviewer 1 Report

This study shows that CPM which has been traditionally used as the medicine in the treatment of thrombocytopenia exerts active effects on platelet production. They seem to show CPM’s action mechanisms through PI3K/Akt and MEK/ERK signaling pathway by appropriate experimental techniques. It seems true that CPM has available clinical profiles.  However I have a major concern about CPM itself. As authors clearly mentioned in Fig.1, CPM has several components in it. I think that it surely has more components unknown too. And there might be some interacting effects of them. I’m wondering if these effects shown here could be induced by any compound in CPM. I hope the authors should discuss about this. There might be some pyrogen substance inducing IL-6. 

1.     How was the dose of CPM prepared for example 20 or 40 ug/mL in treatment of cells or mice? CPM was not solved in water, was it? What was the vehicle in preparation? Was it used in control experiments?

2.     Notes in Fig.4A (Control, CPM (150 mg/kg, Model etc., up of Fig4A) should be ordered as same as others. 

3.     In Fig.7C or 7E, the inhibitors exerted suppression in treatment alone. Do these have a toxic effects on cells?

Author Response

Sep 19, 2022

Dear Expert Reviewer,

Thank you very much for the prompt review process and excellent comments. We greatly appreciate the time and efforts which you have spent on it. We are submitting the revised manuscript entitled “Caulis Polygoni Multiflori accelerates megakaryopoiesis and thrombopoiesis via activating PI3K/Akt and MEK/ERK signal-ing pathways” (ID: 1926270) to Pharmaceuticals.

We have carefully considered your comments and suggestions, and addressed each of the concerns in response to the comments (see point by point response). We have revised the manuscripts based on your comments and carefully checked throughout the manuscript and corrected the language errors. Our point-by-point responses to the comments (in blue) are shown below (in red).

Point 1. This study shows that CPM which has been traditionally used as the medicine in the treatment of thrombocytopenia exerts active effects on platelet production. They seem to show CPM’s action mechanisms through PI3K/Akt and MEK/ERK signaling pathway by appropriate experimental techniques. It seems true that CPM has available clinical profiles. However, I have a major concern about CPM itself. As authors clearly mentioned in Fig.1, CPM has several components in it. I think that it surely has more components unknown too. And there might be some interacting effects of them. I’m wondering if these effects shown here could be induced by any compound in CPM. I hope the authors should discuss about this. There might be some pyrogen substance inducing IL-6.

Response 1: Thanks a lot for the constructive and careful suggestion. We have discussed about the components of CPM and their potential effects and molecular mechanisms on MK differentiation and platelet formation in the discussion section (page 16, line 402-425).

Point 2. How was the dose of CPM prepared for example 20 or 40 µg/mL in treatment of cells or mice? CPM was not solved in water, was it? What was the vehicle in preparation? Was it used in control experiments?

Response 2: At the preliminary screening stage, we used a widely range of drug concentrations to identify the activities of CPM in vitro and in vivo. Pre-experiment results showed that 20, 40 and 80 µg/mL CPM had positive effects on MK differentiation with no cytotoxicity, and 75, 150 and 300 mg/kg CPM had therapeutic effects on RIT mice without system toxicity as manuscript described. The lower concentrations of CPM had no effects on MK differentiation and platelet production, while higher concentrations of CPM could produce toxic effects in vitro and in vivo.  An appropriate amount of CPM was weighed and solubilized in sterilized deionized water/dimethyl sulfoxide (1:1, v/v) for cell treatment. The same volume of vehicle (sterilized deionized water/dimethyl sulfoxide (1:1, v/v)) was added to control group in in vitro experiment. In animal experiments, CPM was dissolved in normal saline and then given to mice by oral gavage. The same volume of normal saline were administered by oral gavage to control and model groups. We have added the description about CPM preparation and administration in materials and methods section (page 18, line 434-437).

Point 3. Notes in Fig.4A (Control, CPM (150 mg/kg, Model etc., up of Fig4A) should be ordered as same as others.  

Response 3: Thank you for your careful reading and excellent suggestion. We have unified the format of label in new Figure 3A.

Point 4. In Fig.7C or 7E, the inhibitors exerted suppression in treatment alone. Do these have a toxic effects on cells?

Response 4: Thank you for your rigorous thinking. Before the inhibitors were used to explore the effect of CPM on MK differentiation, the safe concentration of inhibitors was evaluated by apoptosis and LDH assays in pre-experiments. The results showed that LY294002 (20 μM) and SCH772984 (3 μM) did not induce apoptosis of K562 cells after treatment for 5 days (Figure 1A-D). In addition, the amount of released LDH of K562 cells between control and inhibitors-treated groups had no significant difference (Figure 1E, F). These data demonstrate that LY294002 (20 μM) and SCH772984 (3 μM) have no toxic effects on cells. 

Please find the Figure 1 in the attachment Word file, thanks.

Figure 1. The cytotoxicity of LY294002 and SCH772984 on K562 cells. (A, C) Apoptosis analysis in K562 cells after treatment with LY294002 (20 μM) or SCH772984 (3 μM) for 5 days, respectively. (B, D) Statistical analysis of apoptotic cells (Q2+Q3) in K562 cells after treatment with LY294002 (20 μM) or SCH772984 (3 μM) for 5 days, respectively. (E, F) Detection of LDH activity of K562 cells after treatment with LY294002 (20 μM) or SCH772984 (3 μM) for 5 days, respectively. Data are mean ± SD (n=3, t-test).

Thank you for all the valuable and helpful comments and suggestions. We hope that our revised manuscript is now suitable for publication in Pharmaceuticals.

Best regards,

Jianming Wu

Department of Pharmacology, School of Pharmacy,

Southwest Medical University,

No.1 section 1, Xiang Lin Road, Longmatan District, Luzhou 646000, China

Reviewer 2 Report

In this manuscript, the authors investigated the pharmacological effects of Caulis Polygoni Multiflori (CPM) on thrombocytopenia induced by radiation. This is a well-designed study. The data are robust and straightforward. I have one comment that I would like the authors to address.

• Figure 1 seems to be unnecessary. This analysis can be remover, mentioned in text or moved to a supplementary figure, unless the authors try to identify or discuss a responsible component in CPM.

Author Response

Sep 19, 2022

Dear Expert Reviewer,

Thank you very much for the prompt review process and excellent comments. We greatly appreciate the time and efforts which you have spent on it. We are submitting the revised manuscript entitled “Caulis Polygoni Multiflori accelerates megakaryopoiesis and thrombopoiesis via activating PI3K/Akt and MEK/ERK signal-ing pathways” (ID: 1926270) to Pharmaceuticals.

We have carefully considered your comments and suggestions, and addressed each of the concerns in response to the comments (see point by point response). We have revised the manuscripts based on your comments and carefully checked throughout the manuscript and corrected the language errors. Our point-by-point responses to the comments (in blue) are shown below (in red).

Point 1: Figure 1 seems to be unnecessary. This analysis can be remover, mentioned in text or moved to a supplementary figure, unless the authors try to identify or discuss a responsible component in CPM.

Response 1: Thank you very much for the excellent comments and approval of our work. According to your suggestions, we moved Figure 1 to supplementary figure (Supplementary Figure S1) and the compounds of CPM were discussed in discussion section (page 16, line 402-425).

Thank you for all the valuable and helpful comments and suggestions. We hope that our revised manuscript is now suitable for publication in Pharmaceuticals.

Best regards,

Jianming Wu

Department of Pharmacology, School of Pharmacy,

Southwest Medical University,

No.1 section 1, Xiang Lin Road, Longmatan District, Luzhou 646000, China

Reviewer 3 Report

In this study by Yang et al, the authors have tested the capacity of the Chinese herb Caulis Polygoni Multiflori (CPM) to affect megakaryocyte differentiation, with the prediction that the herb will promote megakaryopoiesis of not only two cell lines but also in mice with radiation-induced thrombocytopenia. The authors first identify chemicals in the herb, and then treat two pro-megakaryocyte (MK) cell lines, K562 and Meg-01 cells, which demonstrated evidence of MK differentiation, albeit with low percentages (some issues with data presentation are mentioned below). Importantly, the cells that show classic MK morphologic characteristics (polypoidy and increased cytoplasmic size) also exhibit modest increases in MK cell surface markers. However, no changes were observed in apoptosis nor a marker of cell damage (LDH release). Most importantly, the authors show thrombocytopenic mice with CPM treatment exhibited increased platelet production, along with increased megakaryocytes in bone marrow and spleen. RNA sequencing analyses identified multiple signaling cascades activated by CPM, results that were confirmed with Western blotting assays on CPM-treated K562 cells, confirming increased protein expression of members in the PI3K/Akt and ERK/MEK signaling pathways. The in vivo data is convincing, and arguably the most important results considering the modest affects shown with the cell lines. There are some issues that need to be considered as follows:

Lines 137-140 and Figs. 1B and C, what is the context of the numbers shown? Without knowing the total number of cells observed, the numbers shown fail to provide an understanding of percentages of cells being affected by CPM, so this must be addressed, including a careful description in the figure legend (no mention of total numbers or field of view is provided in the Materials section). Are the densities consistent in the microphotographs?

Figure 2D, do the authors have control images of either K562 or Meg-01 cells induced to become megakaryocytes? This would be good to show for comparisons to the CPM-treated cells.

Figures 2E-L, again, the authors need to show comparisons to positive controls, such as PMA-induced K562 or Meg-01 cells, which exhibit impressive MK differentiation including polyploidy and increased CD41 expression. The data shown indicates a small increase in MK characteristics, but without any positive control results, this is difficult to judge if the changes are comparable to classic models of MK differentiation. Moreover, did the researchers try to determine if CPM could augment MK differentiation? While the numbers with CPM alone may be small, this could dramatically change with co-induction with other factors known to promote MK differentiation, such as valproic acid, all-trans retinoic acid, etc.

Figure 3E and F, what is the "Control max" values indicating in the figures? This is not explained in the legend, nor is any comparison to LDH levels in the CPM-treated cells.

Lines 311 - 313, the authors should state that the Western blotting results were from treated K562 cells (and perhaps add this to the figure legend).

Figure 7A, under TAL1, five blots are shown yet only 4 conditions are indicated, so this needs to be fixed.

Discussion:

1) Why were the numbers of MK cells so low with the CPM-treated cell lines? This should at least be discussed, as a homogeneous population would be expected to exhibit a more uniform change of cells in response, assuming that the chemicals in the herb are causing significant changes in multiple signaling pathways, as demonstrated by the author's subsequent expression analyses.

2) Do the authors have any insight from the chemical analyses as to which of the identified components might mediate the changes in gene expression, and therefore support of megakaryopoiesis? This should at least be discussed, even if just speculation of specific chemical’s functions on the indicated signaling pathways.

3) Lines 385-393, the listed RNA-Seq analyses are repetitive and thus could be better summarized to indicate multiple types of analyses implicating common pathways affected by CPM, including MAPK, PI3K-Akt, etc.

Materials and Methods:

Lines 459-461, the authors should indicate how CPM was solubilized and prepared for cell line treatments (unless this reviewer missed this explanation). This should also be explained for animal administration (intragastric).

Minor, grammatical issues (note that these are not comprehensive):

Line 61, the phrase "eventually release platelets" is grammatically incorrect - suggest changing to "eventually leading to the release of platelets". 

Line 77, suggest changing "to effective thrombocytopenia treatment...." to "to be effective thrombocytopenia treatments...."

Line 82-86, the sentence has several grammatical errors that need to be addressed.

Line 284, define "DEG" upon first use (assuming differentially expressed genes).

Lines 287-288, add "7" to "Figure C".

Line 324, the sentence is interrupted with a period after "signaling pathways".

Line 352, "has" should be "have" after herbal medicines.

Line 361, "conductive" should be "conducive".

Author Response

Sep 19, 2022

Dear Expert Reviewer,

Thank you very much for the prompt review process and excellent comments. We greatly appreciate the time and efforts which you have spent on it. We are submitting the revised manuscript entitled “Caulis Polygoni Multiflori accelerates megakaryopoiesis and thrombopoiesis via activating PI3K/Akt and MEK/ERK signal-ing pathways” (ID: 1926270) to Pharmaceuticals.

We have carefully considered your comments and suggestions, and addressed each of the concerns in response to the comments (see point by point response). We have revised the manuscripts based on your comments and carefully checked throughout the manuscript and corrected the language errors. Our point-by-point responses to the comments (in blue) are shown below (in red).

Point 1: Lines 137-140 and Figs. 1B and C, what is the context of the numbers shown? Without knowing the total number of cells observed, the numbers shown fail to provide an understanding of percentages of cells being affected by CPM, so this must be addressed, including a careful description in the figure legend (no mention of total numbers or field of view is provided in the Materials section). Are the densities consistent in the microphotographs?

Response 1: Thanks a lot for the constructive and careful suggestion. As you suggest, calculation of amount of big cells without knowing the total number and density of cells is inaccurate. The photographs taken with a microscopy are conducive to display cellular morphology. Therefore, we deleted the statistical graphs of big cells (New Figure 1).

Point 2: Figure 2D, do the authors have control images of either K562 or Meg-01 cells induced to become megakaryocytes? This would be good to show for comparisons to the CPM-treated cells. Figures 2E-L, again, the authors need to show comparisons to positive controls, such as PMA-induced K562 or Meg-01 cells, which exhibit impressive MK differentiation including polyploidy and increased CD41 expression. The data shown indicates a small increase in MK characteristics, but without any positive control results, this is difficult to judge if the changes are comparable to classic models of MK differentiation. Moreover, did the researchers try to determine if CPM could augment MK differentiation? While the numbers with CPM alone may be small, this could dramatically change with co-induction with other factors known to promote MK differentiation, such as valproic acid, all-trans retinoic acid, etc.

Response 2: Thank you for your rigorous thinking. According to your suggestions, we added the data of PMA-induced MK differentiation in revised manuscript (Supplementary Figure S2, page 3, line 131-145). The combination drug therapy is an excellent research point that can assess whether CPM is a promising adjuvant drug for treating thrombocytopenia in clinic. Due to limited time, experimental verification has not been carried out. Many thanks to your valuable suggestions, which will be of great reference value for our subsequent studies.

Please find the Figure S1 in the attchment Word file, thanks.

Figure S1. Megakaryocytic differentiation of K562 and Meg-01 cells induced by PMA. (A) Microscope photographs of K562 and Meg-01 cells with or without PMA treatment (1 nM ) for 5 days were randomly captured at 10 ´ resolution under the inverted light microscope. Scar bar: 100 µm. (B) Giemsa staining of K562 and Meg-01 cells treated with or without PMA (1 nM) for 5 days. Scar bar: 100 µm. (C, E) CD41 and CD42b expression of K562 and Meg-01 cells with or without PMA (1 nM) treatment for 5 days. (D, F) The proportion of CD41+CD42b+ cells in control and PMA positive control group. Data are mean ± SD (n=3, t-test). (G, I) DNA ploidy analysis of K562 and Meg-01 cells with or without PMA (1 nM) treatment for 5 days. (H, J) The percentage of 2N, 4N and ≥ 8N cells in control and PMA positive control group. Data are mean ± SD (n=3, one-way ANOVA). * p < 0.05, ** p < 0.01, *** p < 0.001 vs. the control group.

Point 3: Figure 3E and F, what is the "Control max" values indicating in the figures? This is not explained in the legend, nor is any comparison to LDH levels in the CPM-treated cells.

Response 3: Maximum LDH control was used to assess total amount of LDH present in the cells. It is the positive control of LDH release. We have added the detailed description about LDH assay in materials and methods section (page 19, line 474-482) and given explanation of maximum LDH control in figure legend (page 6, line 167-169).

Point 4: Lines 311 - 313, the authors should state that the Western blotting results were from treated K562 cells (and perhaps add this to the figure legend).

Response 4: We added the description in materials and methods and figure legend sections (page 22, line 611-612; page 15, line 297-299). 

Point 5: Figure 7A, under TAL1, five blots are shown yet only 4 conditions are indicated, so this needs to be fixed.

Response 5: Thank you for your careful reading. The first blot is protein marker and has been deleted (New Figure 6).

Point 6: Why were the numbers of MK cells so low with the CPM-treated cell lines? This should at least be discussed, as a homogeneous population would be expected to exhibit a more uniform change of cells in response, assuming that the chemicals in the herb are causing significant changes in multiple signaling pathways, as demonstrated by the author's subsequent expression analyses.

Response 6: It is known that K562 and Meg-01 cells are classical models for exploring MK differentiation. However, as human erythroleukemic cell line and human megakaryoblastic leukemia cell line, their abilities to differentiate into MK are limited [1, 2]. It may be a reason that CPM has modest effects on MK differentiation of K562 and Meg-01 cells. In the future work, we will establish human CD34+-derived MKs system, which can reproduce the events of normal megakaryopoiesis clearly [3]. We have discussed this point in discussion section (page 15, line 319-326).

Reference:

  1. Alitalo, R., Induced differentiation of K562 leukemia cells: a model for studies of gene expression in early megakaryoblasts. Leukemia research 1990, 14, (6), 501-514.
  2. Schweinfurth, N.; Hohmann, S.; Deuschle, M.; Lederbogen, F.; Schloss, P., Valproic acid and all trans retinoic acid differentially induce megakaryopoiesis and platelet-like particle formation from the megakaryoblastic cell line MEG-01. Platelets 2010, 21, (8), 648-657.
  3. Lanza, F.; Gachet, C.; Eckly, A., In Vitro and In Vivo Methods to Explore Megakaryopoiesis. Journal of visualized experiments: JoVE 2021, (177).

Point 7: Do the authors have any insight from the chemical analyses as to which of the identified components might mediate the changes in gene expression, and therefore support of megakaryopoiesis? This should at least be discussed, even if just speculation of specific chemical’s functions on the indicated signaling pathways.

Response 7: In the 11 compounds of CPM, only nobiletin has been reported that exhibits an ability in inducing MK differentiation through activation of MAPK/ERK-dependent EGR1 expression [4]. There are very few reports about the effects of other compounds on MK differentiation and platelet production. As you suggest, we have discussed the potential relationships between these compounds of CPM and the indicated signaling pathways in discussion section (page 16, line 402-425). In future work, we will identify the active compounds of CPM, elucidate their molecular mechanism in regulating MK differentiation and platelet formation.

Reference:

  1. Yen, J. H.; Lin, C. Y.; Chuang, C. H.; Chin, H. K.; Wu, M. J.; Chen, P. Y., Nobiletin Promotes Megakaryocytic Differentiation through the MAPK/ERK-Dependent EGR1 Expression and Exerts Anti-Leukemic Effects in Human Chronic Myeloid Leukemia (CML) K562 Cells. Cells 2020, 9, (4).

Point 8: Lines 385-393, the listed RNA-Seq analyses are repetitive and thus could be better summarized to indicate multiple types of analyses implicating common pathways affected by CPM, including MAPK, PI3K-Akt, etc.

Response 8: Thank you for your thoughtful comments. We have drawn a brief description about RNA-seq analyses in discussion section (page 15, line 349-351).

Point 9: Lines 459-461, the authors should indicate how CPM was solubilized and prepared for cell line treatments (unless this reviewer missed this explanation). This should also be explained for animal administration (intragastric).

Response 9: We have added the detailed information about CPM preparation and administration for cell line and animal treatments in materials and methods section (page 18, line 433-437).

Point 10: Minor, grammatical issues (note that these are not comprehensive):

Line 61, the phrase "eventually release platelets" is grammatically incorrect - suggest changing to "eventually leading to the release of platelets".

Line 77, suggest changing "to effective thrombocytopenia treatment...." to "to be effective thrombocytopenia treatments...."

Line 82-86, the sentence has several grammatical errors that need to be addressed.

Line 284, define "DEG" upon first use (assuming differentially expressed genes).

Lines 287-288, add "7" to "Figure C".

Line 324, the sentence is interrupted with a period after "signaling pathways".

Line 352, "has" should be "have" after herbal medicines.

Line 361, "conductive" should be "conducive".

Response 10: Thank you for your careful reading and rigorous thinking. We have carefully checked throughout the manuscript and corrected the language and formal errors.

Thank you for all the valuable and helpful comments and suggestions. We hope that our revised manuscript is now suitable for publication in Pharmaceuticals.

Best regards,

Jianming Wu

Department of Pharmacology, School of Pharmacy,

Southwest Medical University,

No.1 section 1, Xiang Lin Road, Longmatan District, Luzhou 646000, China

Round 2

Reviewer 3 Report

The authors have addressed the concerns of this reviewer and improved the impact of the manuscript. The additional comments on possible actions of the identified chemicals in CPM on MEK/ERK or PI3K/Akt signaling pathways is noteworthy, as is the comment that other chemicals in CPM as yet identified could also mediate the shown effects.